# GUIDING LANGUAGE MODEL MATH REASONING WITH PLANNING TOKENS

## ABSTRACT

Large language models (LLMs) have recently attracted considerable interest for their ability to perform complex reasoning tasks, such as chain-of-thought reasoning. However, most of the existing approaches to enhance this ability rely heavily on data-driven methods, while neglecting the structural aspects of the model's reasoning capacity. We find that while LLMs can manage individual reasoning steps well, they struggle with maintaining consistency across an entire reasoning chain. To solve this, we introduce 'planning tokens' at the start of each reasoning step, serving as a guide for the model, and add their embeddings to the model parameters. Our approach requires a negligible increase in trainable parameters (just 0.001%) and can be applied through either full fine-tuning or a more parameter-efficient scheme. We demonstrate our method's effectiveness by applying it to three different LLMs, showing notable accuracy improvements across three math word problem datasets w.r.t. vanilla fine-tuning baselines.

## 1 INTRODUCTION

One of the current challenges for large language models (LLMs) is to solve complex tasks, like math problems. To this end, increasing interest has been devoted to improving the reasoning capabilities of LMs. In this context, the definition of *reasoning* is usually restricted to the auto-regressive generation of intermediary tokens that decompose the input-output problems into a series of steps that make the answer progressively easier to guess, e.g. Chain-Of-Thoughts (CoT) or Program-Of-Thoughts, as popularized by Wei et al. (2022); Wang et al. (2022); Chen et al. (2022).

To improve the reasoning abilities of LLMs, multiple works focused on the production of high-quality reasonings. For example, Yue et al. (2023) fine-tune LLMs on multiple math datasets with CoT and PoT solutions. Yuan et al. (2023) applies rejection sampling on the LLM samples. Other works elicit reasonings from exogenous resources, such as more capable LLMs, i.e. GPT-4 (Mukherjee et al., 2023; Luo et al., 2023). From a high-level perspective, most of these works improve the reasoning abilities of LLMs by creating higher-quality datasets and then simply fine-tuning base LLMs on these datasets (Luo et al., 2023).

Although the quality – and quantity – of reasonings turns out to be crucial, modeling problems still remain due to the length of the reasonings which might lead to compounding of errors during inference. For example, in Figure 1, we show a common error made by Llama2-7B (Touvron et al., 2023) when fine-tuned on the GSM8K dataset (Cobbe et al., 2021). Often, we observe that, for held-out examples, LLMs can generate seemingly correct reasonings but lack global coherence, i.e. the steps progressively drift away from the correct reasoning flow, an observation already made in (Zhang et al., 2023).

In this paper, we tackle this problem by proposing a new fine-tuning technique to guide the generation of reasoning steps. We introduce *planning tokens*, which are special tokens that encode a high-level plan and provide guidance to generate the next step of reasoning, given the previous. Given a dataset of inputs, and corresponding reasonings and answers, we first split each reasoning into a series of steps, i.e. sentences, then prepend a few special planning tokens before each reasoning step. Each planning token can be seen as the verbalization of a *discrete latent variable* that is meant to encode some high-level property of the reasoning step that follows. After the *inference* phase, where we assign planning tokens to each reasoning step in the corpus, the model is fine-tuned

Question: Every day, Wendi feeds each of her chickens three cups of mixed chicken feed, containing seeds, mealworms and vegetables to help keep them healthy. She gives the chickens their feed in three separate meals. In the morning, she gives her flock of chickens 15 cups of feed. In the afternoon, she gives her chickens another 25 cups of feed. **How many cups of feed does she need to give her chickens in the final meal of the day** if the size of Wendi's flock is 20 chickens?

| **Vanilla supervised fine-tuning** | **Fine-tuning with arithmetic planning tokens** |
|---|---|
| Every day, Wendi gives her chickens 15 cups of feed in the morning + 25 cups of feed in the afternoon = <<15+25=40>>40 cups of feed. | \<prefix\> \<+\> Wendi gives her flock 15 cups of feed in the morning and another 25 cups in the afternoon, for a total of 15+25 = <<15+25=40>>40 cups of feed. |
| If she has 20 chickens and she feeds them 40 cups of feed every day, then each chicken gets 40/20 = <<40/20=2>>2 cups of feed per chicken. | \<prefix\> \<*\> If Wendi has 20 chickens, then she needs 20*3 = <<20*3=60>>60 cups of feed to feed her flock. |
| The answer is: 2    *Drifting away from the correct reasoning flow* | \<prefix\> \<-\> If Wendi has already given her flock 40 cups of feed, then she needs to give her flock 60-40 = <<60-40=20>>20 more cups of feed. |
| | \<prefix\> \<answer\> The answer is: 20 |

Figure 1: An example of a question from the GSM8K dataset, along with two chain-of-thoughts solutions generated by LLMs. **Left**: The reasoning chain generated by vanilla supervised fine-tuning LLM drifts away from correct reasoning flow and leads to a wrong answer. **Right**: we use planning tokens to guide an LLM's generation at every reasoning step to encourage the correct reasoning flow. For simplicity, we show the variant of our method which utilizes arithmetic planning tokens: `<prefix>`, `<+>`, `<*>`, `<->`, `<answer>` are the planning tokens added to the original vocabulary.

on the resulting augmented dataset to first predict each planning token, and then the corresponding reasoning step. Crucially, each planning token is associated with a learnable embedding that is learned end-to-end during fine-tuning. In practice, the planning tokens add negligible additional learnable parameters to the LLMs embedding matrices ($< 0.001\%$).

We study increasingly complex ways of inferring the planning tokens for each reasoning step, ranging from designing simple rules to learning a neural clustering model that operates in sentence embedding space. From a higher-level probabilistic perspective, our method can be seen as one iteration of an expectation-maximization algorithm: first, we fill in the missing values for the discrete latent variables in the dataset; then we learn the parameters of the underlying probabilistic model by fine-tuning the LLM on the complete dataset.

Our technique is orthogonal to approaches that create new reasoning datasets. Our method can also be combined with parameter-efficient fine-tuning techniques such as LoRA (Hu et al., 2021). We show that the gains of our method do not solely come from the augmented computational space induced by increasing the length of the reasoning with planning tokens but also from the specialization induced by the planning tokens. Overall, our method improves upon the baseline without planning tokens by 3.3% accuracy points on average over three pre-trained language models (Phi 1.5, 7B Llama 2, and 13B Llama 2) and three math word datasets. The promising empirical results shed light on exciting future works in different directions.

## 2 METHOD

**Setup** We assume that we have a dataset $\mathcal{D}$ composed of triples $\{(x, r, y)\}$, where $x$ is a problem to solve, $r$ is the ground-truth reasoning associated with the problem and $y$ is the final answer. All of $x$, $r$, and $y$ are expressed in natural language and can be tokenized by an LLM. The assumption that the ground-truth reasoning is included in the dataset might be relaxed by either generating CoTs with rejection sampling from a base LM (Yuan et al., 2023) or by creating reasoning paths from a larger model (Luo et al., 2023; Mukherjee et al., 2023).

One of the straightforward ways to leverage this type of data for fine-tuning an LLM is to concatenate each tuple $(x, r, y)$ into a single sequence and fine-tune a base LLM to predict the concatenation of reasoning and answers given the inputs, i.e.:

$$\max_\theta \sum_\mathcal{D} \log p(r, y|x; \theta), \tag{1}$$

where we assume that $x, r, y$ are properly tokenized thus the loss is next-token prediction and $\theta$ is the parameter of the pre-trained LLM. This strategy has been studied and leveraged in many recent efforts (Kim et al., 2023; Yue et al., 2023). We also assume that the reasoning $r$ can be semantically split into multiple steps $r^1, \ldots, r^N$, where $N$ is the number of steps in reasoning $r$.

## 2.1 PLANNING TOKENS AS DISCRETE LATENT VARIABLES

We adopt a Bayesian view of language models, which means we assume there is a latent variable $\kappa$ governing the text generation process:

$$p(w_{n+1}|w_{1:n}) = \int_K p(w_{n+1}|\kappa)p(\kappa|w_{1:n})d\kappa$$

where $w_i$ denotes a token in the text sequence. We assume there exists an optimal value of $\kappa = \kappa^*$ for each step of generation, such that $\mathrm{argmax}_w p(w|\kappa^*)$ is the Bayes optimal predictor (i.e. has the lowest cross-entropy loss) for the next token $w_{n+1}$. Note that $\kappa$ can be quite complex and hard to model in general, considering the richness of text expressions. However, in the scenario of chain-of-thoughts reasoning, the structure of $\kappa$ becomes clearer: the distribution of $\kappa$ should reflect the coherence within each step, and large changes should only happen at the beginning of a new step.

Here, we make a simplified assumption that the discrete planning variable $t \in \{1 \ldots P\}$ is a proxy of $\kappa$. We also assume that the distribution of $\kappa$ does not change much within each step, while it can change drastically at the beginning of a new step. So we only need to re-estimate $t$ at the beginning of each reasoning step. Ideally, the next reasoning step can be solely generated by $\kappa$. However, since we do not have the ground truth $\kappa$ and only have the planning variable $t$, we need to utilize all available information, including all the previous reasoning steps and their corresponding $t$'s, to infer the current step.[1] By conditioning on the proxy planning variable $t$, we can better control the generation by getting closer to the optimal $\kappa = \kappa^*$:

$$p(r^{n+1}|x, t^1, r^1, \ldots, t^n, r^n, t^{n+1}) = \int_K p(r^{n+1}|\kappa)p(\kappa|x, t^1, r^1, \ldots, t^n, r^n, t^{n+1})d\kappa$$

Since $p(\kappa|x, t^1, r^1, \ldots, t^n, r^n, t^{n+1})$ will be more concentrated on $\kappa^*$ than without planning tokens $p(\kappa|x, r^1, \ldots, r^n)$, the resulting marginal distribution $p(r^{n+1}|x, t^1, r^1, \ldots, t^n, r^n, t^{n+1})$ will be closer to the Bayes optimal classifier $p(r^{n+1}|\kappa^*)$ than without planning tokens $p(r^{n+1}|x, r^1, \ldots, r^n)$. The planning token can also be considered as condensed summaries of each reasoning step and can therefore provide useful guidance while producing the reasoning $r$. We also assume the answer $y$ is generated with a special planning variable $t^y$ which has always a value equal to $P$ for each example. Without loss of generality, we assume that only $t^y$ can take value $P$.

If the planning variables were indeed observed, the complete data $\mathcal{D}^c$ would be composed of tuples $(x, t^1, r^1, \ldots, t^S, r^S, t^y, y)$. Let us assume for the moment that the complete dataset is available to us. In order for the dataset to be processed by an LLM, we verbalize each planning variable $t^i$ with a *planning token*, which we will also denote as $t^i$ for simplicity. In practice, this means that we extend the vocabulary of the LLM tokenizer with $P$ (or more) new tokens of our choice, and modify accordingly the input and output embedding matrix of the LLM. The embeddings of the new tokens will be additional training parameters for the LLM, which add negligible parameter overhead over the base model (i.e., they add $\leq 0.001\%$ of the total number of parameters).

With this modification in mind, the complete data $\mathcal{D}^c$ can be readily used to train an LLM by maximizing the complete data log-likelihood:

$$\max_{\theta^+} \sum_\mathcal{D}^c \log p(t^1, r^1, \ldots, t^S, r^S, t^y, y|x; \theta^+) \tag{2}$$

---

[1] We have tried to reduce the full conditioning by enforcing a sparse attention scheme to ensure $t^i$ only conditions on $x$ and $t^1, t^2, \ldots, t^{i-1}$, and $r^i$ only conditions on $t^i$, which results in a large performance drop.

where by $\theta^+$ we denote the original LLM parameters comprising the embeddings of the planning tokens. From the equation above, we can see that the LLM will be trained to first generate the high-level planning token for each reasoning step, and then generate the detailed reasoning step. Intuitively, this can help LLMs generate more consistent and controlled reasoning chains.

## 2.2 INFERRING PLANNING TOKENS WITH PLANNING TYPES

Imputing the values of the latent variables to create the complete dataset remains a big hurdle. Ideally, we would want to recur to expectation-maximization, or variational EM, to learn an approximate posterior $q(t^i|x, r^1, r^2, ..., r^i)$, and parameters $\theta^+$, to maximize (a lower-bound on) the marginal log-likelihood of the observed data (Equation (2)). This problem might reveal itself to be difficult, due to 1) the number of latent variables involved in the computation; 2) the discrete nature of the latent variables which makes gradient-based optimization difficult; and 3) the sheer number of parameters involved in maximization w.r.t. $\theta^+$. Therefore, we focus on examining whether simpler heuristics could prove valuable as approximations for posterior inferences. We loosely refer to these strategies as *planning types*. Note that these inferences are only done at training time with training data, while at test time the value of the planning tokens is predicted using the LLM itself.

### 2.2.1 ARITHMETIC OPERATORS

As we are focusing on math word problems in this paper, it is natural to consider the basic arithmetic operation contained in each reasoning step $r^i$ as the plan token $t^i$ similar to Zhang et al. (2023). This yields four planning tokens ($+, -, \times, \div$), which can be easily extracted from each of $r^i$. Therefore, in this setting, the LLM will be trained to first commit to a particular type of operation, and then to generate the reasoning that verbalizes the operation. If more than one operator is found, then we use as many planning tokens as the number of operations found, e.g. `<+><*>`.

This approach has several drawbacks. First, some operations, such as doubling a quantity, can be implemented with more than one operator (e.g. + and $\times$). Thus, this arithmetic partitioning can fail to capture the similarity between these reasoning steps. Second, for some datasets such as MATH, such heuristics might not be simple to design. Finally, the number of planning types is invariant to the number of examples in the dataset, which might be a too-strict constraint. To relax these shortcomings, we turn to clustering-based approaches.

### 2.2.2 CLUSTERING-BASED APPROACHES

In clustering-based approaches, we first gather reasoning steps $r^i$ for all the reasonings $r$ in our dataset, then apply a clustering algorithm on the representations $\phi(r^i), \forall r^i \in \mathcal{D}$. We obtain $\phi(r^i)$ by using the base LLM model to encode the whole text sequence of an example $(x, y)$, and then averaging the last hidden layer outputs of the corresponding tokens in $r^i$. In this way, we aggregate both the question and previous steps' information in the contextualized representation $\phi(r^i)$. Once we obtained the contextualized embeddings $\{e^i = \phi(r^i)\}_i$ for all training data, we experimented with two types of clustering algorithms: K-Means[2] and a VAE-based non-linear clustering method.

**Linear Clustering: K-Means** We simply run an off-the-shelf K-Means implementations on the set of representations $e^i$ for all reasoning steps in the dataset $\mathcal{D}$. Each planning variable $t^i$ is then assigned the index of the centroid closest to $r^i$.

**Non-Linear Clustering: Soft Q-VAE** In this approach, our aim is to learn a more expressive non-linear transformation of the embedding $e^i$ that echoes our discrete latent variable definition of the planning tokens. We utilize a variational autoencoder (VAE) (Kingma & Welling, 2014) with a Softmax-like soft quantization of the latent space (Miao et al., 2017; Potapczynski et al., 2020) to implement our probabilistic assumptions.[3] Our VAE maximizes the following objective derived from ELBO (Kingma & Welling, 2014):

$$\mathcal{L}(\mu, \sigma, f, g) = \mathbb{E}_{z \sim \mathcal{N}(\mu(e^i), \sigma(e^i)^2)}[(f(g(z)) - e^i)^2] - \text{KL}[\mathcal{N}(\mu(e^i), \sigma(e^i)^2)||\mathcal{N}(0, I)], \quad (3)$$

---

[2]We also experimented with a regularized version of K-Means to ensure a more balanced allocation across clusters, but did not see any significant improvements.

[3]We chose a soft version of quantized VAE instead of the classic VQ-VAE (Oord et al., 2017) because we found VQ-VAE is hard to train on our embedding data.

The first log-likelihood term can be reduced to squared error because the distribution of $e^i$ is assumed to be Gaussian. The KL-term can be reduced to Gaussian because $g$ is an invertible soft quantization function, that relaxes the discrete latent planning variable $t \in \mathcal{S} = \{1, 2, ..., P\}$ to a continuous space $g(z) \in \mathcal{S}^{(P-1)} = \{x \in \mathbb{R}^{K-1} : x_k > 0, \sum_{k=1}^{P-1} x_k < 1\}$, where $z$ is a Gaussian variable. According to Potapczynski et al. (2020), the most natural choice of $g$ is a modified Softmax function:

$$g(z, \tau)_k = \frac{\exp(z_k/\tau)}{\sum_{j=1}^{K-1} \exp(z_j/\tau) + \delta}, \ \forall k \leq P - 1$$

where $\tau > 0$ is the temperature and the constant $\delta > 0$ is to make sure $g$ is invertable. We find the approximated discrete value of $t^i$ by $t^i = \text{argmax}_k g(\mu(e^i), 0)_k$.

### 2.3 PLANNING TOKENS PARAMETRIZATION

Each planning variable can be verbalized by one or more planning tokens. Moreover, we discussed two orthogonal effects that might be achieved by augmenting the dataset with planning tokens. The first is to augment the computational capacity of the model by providing additional "scratch" space to predict the next reasoning step; the second is the specialization induced by information-bearing planning tokens. To be able to disentangle both effects in our model, we introduce two hyper-parameters in our approach: *n_prefix* and *n_special*. Each planning variable can be verbalized using a variable number of both generic *prefix* planning tokens and *special* planning tokens. For example, the configuration illustrated in Figure 1 (right) has been achieved by setting *n_prefix* = *n_special* = 1, which means that we add 1 generic planning token `<prefix>` *independent* of the value of the planning variable and 1 special planning token depending on its value, e.g. `<+>`. If we augment *n_special*, we create new planning tokens as follows `<+_1>`, whose embeddings are included in $\theta^+$ as additional parameters to be learned. For the clustering-based approaches, we create special planning tokens such as `<k-means_1>` and `<vae_1>`.

## 3 EXPERIMENTS

### 3.1 EXPERIMENTAL SETUP

**Datasets** – Our core analysis relies on three mathematical reasoning datasets, each containing detailed reasoning steps. The Grade School Math dataset (GSM8K) (Cobbe et al., 2021) contains 8.5K examples of linguistically diverse grade school math world problems. Next, we use the MATH dataset (Hendrycks et al., 2021), a collection of 12.5K challenging competition mathematics problems. Lastly, we also use the AQUA-RAT dataset (Ling et al., 2017) containing 100K samples of mathematical problems, along with sequences of human-readable mathematical expressions in natural language. We randomly sample a 1K subset of the original test set of GSM8K and MATH to test our models for compute efficiency. We use the original test set of AQUA it is already under 1K.

**Base LLMs** – Our empirical analysis uses several decoder-only architectures of varying sizes. We use the 7B and 13B variants of Llama2 (Touvron et al., 2023), both trained over 2 trillion tokens from publicly accessible data sources and with a context window of 4096 tokens. We also experiment with Phi-1.5 (Gunasekar et al., 2023), a 1.3B parameter model trained on a mixture of textbook-quality code data, and additional synthetically generated textbook and exercise data.

**Training** – Due to excessive computational requirements, in the experiments using Llama2 as the base model, we rely on low-rank adapters (LoRAs) (Hu et al., 2021) to fine-tune the base LLM. We apply LoRAs to the projection parameters of each MLP block in the base LLM. For all methods, we use a rank of 16, and a dropout of 0.05. To further reduce the memory usage, we load 13B Llama2 in 8 bits at training time. For Phi-1.5, we perform full-model fine-tuning. For our variants with planning tokens, we add the embeddings of the planning tokens to the learnable parameters of the base model, which results in an increase of less than 0.001% of the total trainable parameters. We train all models for 10 epochs, with a learning rate of 2e-5 using the AdaFactor optimizer (Shazeer & Stern, 2018) for full fine-tuning, and a learning rate of 2e-4 using the AdamW optimizer (Loshchilov & Hutter, 2017) for parameter efficient tuning.

**Baselines and Methods** – Our models are denoted with the three different planning types: **Arithmetic**, **K-Means** and **SQ-VAE**. We compare our models to several baselines. First, we compute the

| Base model | Planning Type | #clusters | #trainable | GSM8K | MATH | AQUA | Avg |
|---|---|---|---|---|---|---|---|
| Phi 1.5 | N/A | 0 | 100% | 12.5 | 1.3 | 27.2 | 13.5 |
| (1.3B) | General | 1 | 100% | 15.4 | 2.0 | 35.4 | 17.6 |
| | **Arithmetic** | 4 | 100% | 15.0 | 2.3 | 33.1 | 16.8 |
| | **K-Means** | 5 | 100% | 14.5 | 2.7 | **36.5** | 17.7 |
| | **SQ-VAE** | 5 | 100% | **15.8** | **3.3** | 34.3 | **17.8** |
| Llama2 | Prefix | 1 | 0.023% | 8.9 | 3.6 | 32.9 | 15.1 |
| (7B) | Prompt | 1 | 0.001% | 15.2 | 5.3 | 26.8 | 15.8 |
| | N/A | 0 | 0.343% | 38.2 | 6.5 | 36.6 | 27.1 |
| | General | 1 | 0.344% | 38.5 | 6.7 | 37.8 | 27.7 |
| | **Arithmetic** | 4 | 0.344% | 39.5 | 5.6 | 38.2 | 27.8 |
| | **K-Means** | 5 | 0.344% | 39.1 | 6.7 | 40.5 | 28.8 |
| | **SQ-VAE** | 5 | 0.344% | **40.0** | **7.0** | **41.3** | **29.4** |
| Llama2 | Prefix | 1 | 0.019% | 16.0 | 3.2 | 30.3 | 16.5 |
| (13B) | Prompt | 1 | 0.001% | 27.8 | 6.4 | 26.0 | 20.1 |
| | N/A | 0 | 0.279% | 44.6 | 7.2 | 41.3 | 31.0 |
| | General | 1 | 0.280% | 47.9 | 7.9 | 42.5 | 32.8 |
| | **Arithmetic** | 4 | 0.280% | 41.9 | 4.6 | 35.8 | 27.4 |
| | **K-Means** | 5 | 0.280% | 49.6 | 8.4 | **44.1** | 34.0 |
| | **SQ-VAE** | 5 | 0.280% | **50.6** | **8.5** | 43.9 | **34.3** |

Table 1: Testing accuracy of fine-tuned language models on different math word datasets. We set the number of planning tokens ($n\_prefix+n\_special$) to 6.

performance of the full-finetuning and LoRA-based adaptation in the case of Phi-1.5 and Llama-2, respectively (**N/A**). Then, we ablate whether the benefit of using planning tokens comes solely from increasing the computational space for the model (**General**). To do so we set $P = 1$, i.e. all planning variables take the same value. This baseline benefits from augmented computational capacity given by inserting additional tokens in the sequence but does not provide any specialization to the ground-truth next reasoning step. We also compare with the classic **Prefix** (Li & Liang, 2021) and **Prompt** (Lester et al., 2021) tuning baselines. Note that we use the same number of prompt tokens (6) as our method since adding more tokens to match the number of trainable parameters does not improve their performance.

## 3.2 RESULTS

We present our main results in Table 1. Generally, we observe that for all three datasets considered and all the model sizes, the best-performing approach leverages planning tokens. We note that, across scales, **General** improves over vanilla fine-tuning (**N/A**), indicating that adding additional learnable tokens before each reasoning step gives the model additional computation space and results in better performance. For the three datasets, **Arithmetic** does not seem to consistently outperform **General** pointing to the fact that hand-designed heuristics might not be optimal and the gain observed over **N/A** might just be due to additional allowed computation space. However, our clustering methods that infer planning tokens consistently outperform both **General** and **Arithmetic**, pointing to the importance of the inference step and planning tokens specialization. Over the 7B version of Llama2, all methods with planning tokens provide consistent gains over the baselines of up to 1.7 points, a relative increase of 6%. Our results over the Llama2 13B tell a similar story, where again the best-performing method across all datasets uses planning tokens. We show a 1.5 gain in average accuracy (4.5 % relative gain). Note that the classic soft-prompt-tuning-based methods (**Prefix**, **Prompt**) do not work on our datasets likely due to their complexity. That is, only modifying the prefix space of a sequence is not adequate for solving math reasoning problems. Changes in the internal parameters and guidance in between the steps are necessary.

**Ablation** We show ablation studies with respect to the number of clusters $P$ and the number of planning tokens used in total. A larger $P$ means we classify reasoning steps into more fine-grained planning types. Note that $P = 1$ means we always use the same planning tokens for all steps so it is the same as the **General** baseline. From the left table of Table 2, We observe that the performance of both **K-Means** and **SQ-VAE** first goes up and then goes down when the number of clusters

| Plan Type | Number of Clusters $P$ | | | | |
|---|---|---|---|---|---|
| | 1 | 3 | 5 | 7 | 10 |
| General | 38.5 | - | - | - | - |
| K-Means | - | 37.9 | 39.1 | 39.9 | 36.7 |
| SQ-VAE | - | 39.6 | 40.0 | 39.4 | 38.9 |

| Plan Type | Number of Plan Tokens | | |
|---|---|---|---|
| | 2 | 6 | 10 |
| General | 37.9 | 38.5 | 38.3 |
| Arithmetic | 39.5 | 38.9 | 38.5 |
| K-Means | 38.9 | 39.1 | 38.1 |
| SQ-VAE | 40.0 | 38.8 | 38.2 |

Table 2: Impact of varying the number of clusters (**left**) and the number of planning tokens (*n_prefix+n_special*, **right**). For general, we do not have any special tokens, so we match computation by adding prefix tokens.

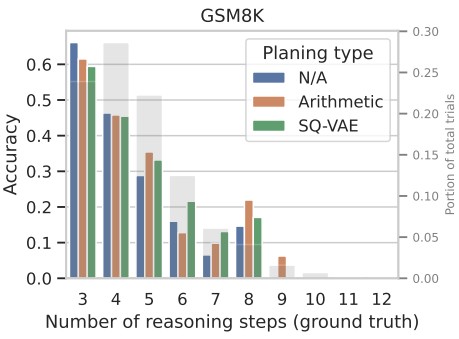 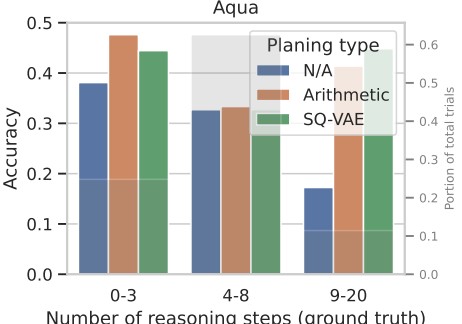

Figure 2: Accuracy on GSM8K (**left**) and Aqua (**right**) on test examples by their number of ground-truth reasoning steps. SQ-VAE consistently increases performance for test examples that require more steps of reasoning to be solved.

increases. This likely reflects that our planning token inference models are not able to produce too fine-grained reasoning type. From the right table of Table 2, we observe that it is not necessary to assign too many tokens to a planning type. We suspect that too many planning tokens will result in a longer sequence which negatively affects the language model's reasoning ability.

## 3.3 ANALYSIS

**Errors by Reasoning Length** In Figure 2, we show the accuracy by the complexity of the test example as measured by reasoning length on AQUA and GSM8K. For the two datasets, **SQ-VAE** outperforms the baseline especially for examples requiring longer reasonings, thus hinting that the planning tokens provide better control over long reasoning chains. For GSM8K, planning tokens seem to underperform the baseline for short reasoning chains. In the future, we might want to explore position-wise specialization of the planning tokens, i.e. use a different variant of the planning token w.r.t. the position of the step in the reasoning.

**Error Taxonomy** After inspecting the generated chain-of-thoughts solutions from all three datasets, we propose to classify the errors into the following five categories:

- **Misunderstanding of question**: the generated solution does not answer the given question.
- **Computation errors**: errors in evaluating math equations.
- **Inaccurate extraction of question information**: references to the information provided in the question is wrong in the generation.
- **Wrong application of math knowledge**: the equation, concept, or theorem used in the generated solution is not suitable for the question.
- **Wrong logic/unreasonable step**: the generated step does not follow common logic.

We prompt GPT4 to obtain the error type of each generated example, and then manually verify whether the predicted error type matches the explanation. In Figure 3, the **SQ-VAE** error frequencies significantly lower than the **N/A** baseline are in bold font. As we can see, across three datasets, our method consistently improves on the computation errors and the inaccurate extraction of question

| Error type | Misunderstanding of question | | Computation errors | | Inaccurate extraction of question information | | Wrong application of math knowledge | | Wrong logic | |
|---|---|---|---|---|---|---|---|---|---|---|
| Example | Question: ...How many cups of feed does she need to give her chickens in the final meal of the day... ...then each chicken gets 40/20 = <<40/20=2>>2 cups of feed per chicken. The answer is: 2 | | ...7 + 3301x = 3371x... | | Question: ...He spends the next half-hour driving at a speed of 30mph... ...He drove 2 hours at 30mph so he traveled 2*30=<<2*30=60>>60 miles... | | ...The number of feet the plane is from the ground is the target of a geometric sequence... | | ...$2ab = 12 = 2^3 \cdot b^2$... | |
| Dataset | N/A | SQ-VAE | N/A | SQ-VAE | N/A | SQ-VAE | N/A | SQ-VAE | N/A | SQ-VAE |
| GSM8K | 4.1% | 4.1% | 8.0% | **6.1%** | 34.5% | **33.0%** | 0.2% | 0.5% | 16.2% | **14.9%** |
| MATH | 21.0% | 23.5% | 8.1% | **5.0%** | 27.3% | **24.7%** | 20.3% | **18.2%** | 23.1% | 23.0% |
| AQUA | 8.3% | 8.7% | 12.6% | **9.4%** | 25.6% | **23.2%** | 2.8% | 2.0% | 19.5% | 20.1% |

Figure 3: Example of different error types. The frequency of each error type generated by the **N/A** baseline and our **SQ-VAE** planning method on test sets are predicted by GPT4 and then manually verified.

information. The lower computation errors indicate that our inferred planning tokens either increase the computation capacity of the model or contain compute-related information. Our method being able to make more faithful reference to the information in the questions indicates planning tokens are helpful for long-range control of generations, which echoes the observation and assumptions in the previous sections. The differences on other error categories are either insignificant or inconsistent over datasets. Note that GSM8K and AQUA have a similar distribution over the error types, while MATH has significantly more misunderstanding of questions and wrong application of math knowledge. This is understandable as MATH is significantly more difficult as it is from American Mathematics Competitions (AMC). These questions are less obvious to approach and require applications of non-trivial math theorems.

**Distinguishability of the Induced Clusterings** In this section, we investigate whether the categorization of reasoning steps sentences, categorized by different planning tokens, has captured useful information to distinguish those sentences. We use a probing task Alain & Bengio (2017), framed as a multi-class classification problem. Concretely, we collect a dataset consisting of a set of sentences and their planning token labels obtained from a certain planning type (e.g., **K-Means**, **SQ-VAE**). We train a simple model (either a logistic regression or a shallow neural network) to learn the mapping from a sentence to its planning token label. We hypothesize that the categorization performed by better planning type should facilitate easier learning dynamics for simple models because the class boundaries should be easily accessible from the data.

As a case study, we compare between **K-Means** and **SQ-VAE**, the two stronger planning types reported in our main results, we study the 5-cluster setting on the GSM8K dataset. Given a planning type, we randomly sample 2000 sentences from each of the five categories, resulting in 10K sentences and their corresponding class labels. We randomly split the data into 50%/25%/25% as training, validation, and test sets. We leverage a pre-trained text encoder[4] to convert sentences into vector representations. Taking the text encodings as input, the probe network (logistic regression, 2-layer MLP, or 6-layer MLP)

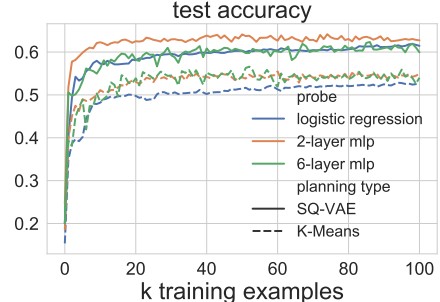

Figure 4: Test accuracy of the probes on the sentence classification task.

performs a 5-way classification. The task of the probe network is to learn a function such that it minimizes the cross-entropy loss against ground-truth labels.

We show test accuracy curves in Figure 4. We observe a clear gap between **SQ-VAE** (solid lines) and **K-Means** (dashed lines) in the curves. This reflects our main results reported above, where **SQ-VAE** produces a better overall downstream task performance, it also to some extent verifies our hypothesis that a better planning type could provide a sentence categorization that is easier to access by shallow classifiers.

---

[4]We use **mpnet-base** Song et al. (2020), a default model in Sentence-Transformers Reimers & Gurevych (2019).

## 4 RELATED WORK

**Reasoning** is the ability of LMs to analyze, infer, and make logical connections between pieces of information to arrive at a meaningful new conclusion (Huang & Chang, 2022). One of the most influential techniques to facilitate reasoning in LMs is Chain-of Thought (CoT), where the LM is either prompted (Wei et al., 2022) or trained (Kim et al., 2023) to generate reasoning steps before giving the final answer. Several recent extensions of CoT include training versifiers to rank reasoning chains (Cobbe et al., 2021; Li et al., 2023), enforcing self-consistency (Wang et al., 2022), organizing reasoning steps in trees/grpahs instead of chains (Yao et al., 2023; Besta et al., 2023). In this work, we focus on evaluating the reasoning ability of LMs through the lance of math word problems (MWP).

**MWPs** typically mean a word-described question requiring mathematical operations for a solution. Approaches to solving MWPs can be roughly divided into three groups: (1) symbolic reasoning approaches: convert text into symbols with rules and executing math operations directly on these symbols (Bobrow, 1964; Fletcher, 1985); (2) modern approaches that employ DNNs to directly predict the answer (Wang et al., 2018; 2017; Shen et al., 2021; Lewkowycz et al., 2022); (3) lastly, and with particular relevance to this work, CoT based approaches (Zhang et al., 2023; Li et al., 2023) that train solvers to generate reasoning steps before giving the final answer. Specifically, Zhang et al. (2023) performs CoT fine-tuning of GPT2s by first predicting the math operation of each reasoning step at generation time, which is significantly less efficient than our method.

**LM fine-tuning** methods aim to adapt underlying base LM to specific domains (Cheng et al., 2023), or endow them with specific skills (Mishra et al., 2021). Several fine-tuning datasets (Lee et al., 2023; Longpre et al., 2023) and dataset generation techniques (Xu et al., 2023) have been proposed, including the math domains (Luo et al., 2023). Applying planning tokens to domains other than math is a promising avenue for future research. **Parameter efficient fine-tuning** (PEFT) reduces memory and time of LM fine-tuning through training a small number of infused parameters at selected layers (Houlsby et al., 2019; Ansell et al., 2021) while freezing the backbone. Some of the popular PEFT approaches include LORA (Hu et al., 2021), which infuses low-rank weights, IA3 (Liu et al., 2022), which scales the activations. Many earlier PEFT works tune the embeddings of the infused new prefix tokens (Qin & Eisner, 2021; Li & Liang, 2021; Lester et al., 2021), which are not expressive enough for MWPs as shown in our experiments.

**Clustering** information can be leveraged for more effective training of LMs, such as determining text domains (Aharoni & Goldberg, 2020) for mixture-of-experts training (Gururangan et al., 2023; Gross et al., 2017; Chronopoulou et al., 2023; 2021; Duan et al., 2021) or pseudo-labeling for effective unsupervised learning (Caron et al., 2018). Various text clustering techniques have demonstrated their effectiveness on textual data including clustering on top of tf-idf embeddings (Gururangan et al., 2023), neural embeddings (Chronopoulou et al., 2023), using k-means clustering based on balanced linear assignment problem (Malinen & Fränti, 2014), soft clustering with GMMs (Chronopoulou et al., 2023) or topic-modeling flavours (Duan et al., 2021). In this work, we leveraged clustering for the planning token assignment to reasoning steps. More specifically, we use k-means clustering on top of step embeddings, and clusters induced by a latent variable model.

## 5 CONCLUSION

We proposed to use planning tokens to gain better control over reasonings with LLMs. Our planning tokens increase the performance over baselines at a minimal parameter cost. We studied our method's efficacy across three different base models and multiple math word problem datasets.

First, we see a straightforward extension of our framework which could use multiple planning variables at every reasoning step, each planning variable is responsible for deliberating in a different "view" of that particular reasoning step (Wang et al., 2023). Second, future work should go beyond our heuristic inference procedures and learn the inference network, such as to maximize the marginal log-likelihood of the observed data: we could then interpret the overall model as a Sequential VAE (Goyal et al., 2017). Finally, it is meaningful to continue the exploration towards interpretability and explainability of the planning tokens (Khashabi et al., 2021). We believe such research could shed light on prompt searching/optimization studies performed both by humans (Zamfirescu-Pereira et al., 2023) and machines (Shin et al., 2020; Sordoni et al., 2023).

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

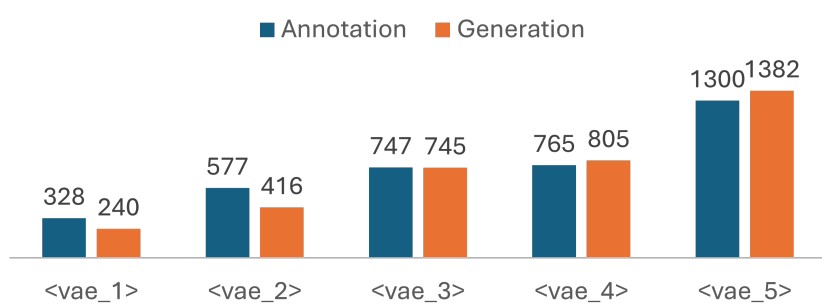

Figure 5: The marginal distribution (count) of the SQ-VAE planning tokens over GSM8K test set.

## A APPENDIX

**Planning token distribution**

In Figure 5, we show the frequency of each planning tokens appears in the GSM8K test set. Annotation means the planning type predicted by the SQ-VAE. Generation means the planning token generated by the fine-tuned language model. As we can see, the marginal distribution of the planning variable approximately matches between SQ-VAE and language model, which means fine tuning language model with planning tokens can effectively learn to infer the correct planning type from the previous steps.

**GPT4 prompting details for error type classification**

Here is the prompt we used for error type classification:

```
Your goal is to identify the error type in reasoning steps of an AI agent
    in response to a question.

You are given the corresponding question, the reasoning steps generated
    by an AI model and the ground truth reasoning steps in the format:

####Question: [question]

####Generated Answer: [reasoning]

####GT reasoning: [ground truth reasoning].

Your task is to identify the error type in AI's reasoning. There are 5
    different error types: (1) Misunderstanding of question, (2)
    Computation errors, (3) Inaccurate extraction of question information
    , (4) Wrong application of math knowledge, (5) Wrong logic/
    unreasonable step.

Here are some examples of each of the error types:

1 Misunderstanding of question: ask about total number of cups in the
    final meal, answer about number of cups each chicken get.: Question:
    Every day, Wendi feeds each of her chickens three cups of mixed
    chicken feed, containing seeds, mealworms and vegetables to help keep
     them healthy. She gives the chickens their feed in three separate
    meals. In the morning, she gives her flock of chickens 15 cups of
    feed. In the afternoon, she gives her chickens another 25 cups of
    feed. How many cups of feed does she need to give her chickens in the
     final meal of the day if the size of Wendi's flock is 20 chickens?
    Every day, Wendi gives her chickens 15 cups of feed in the morning +
    25 cups of feed in the afternoon = <<15+25=40>>40 cups of feed. If
    she has 20 chickens and she feeds them 40 cups of feed every day,
    then each chicken gets 40/20 = <<40/20=2>>2 cups of feed per chicken.
     The answer is: 2
```

2 Computation errors: 7 + 3301x != 3371x: Question: What is the smallest positive integer $N$ such that the value $7 + (30 \times N)$ is not a prime number? We have $7 + (30 \times N) = 7 + 30 + 300 + 3000 + \dots + (30) \times N = 7 + 3301x = 3371x$ We write $3371x$ as $3300 +71x$, in order to factor out a multiple of 33 We can try adding one to $71$ and subtracting it from $3300$ If we add 2, we get $73$, but $73$ is prime If we add 4, we get $75$, but $75$ is prime If we add 8, we get $83$, but $83$ is prime If we add 16, we get $99$, but $99$ is prime If we add 32, we get $131$, but $131$ is prime If we add 64, we get $195$, but $195$ is prime If we add 128, we get $323$, but $323$ is prime If we add 256, we get $581$, but $581$ is prime If we add 512, we get $803$, but $803$ is prime If we add 1024, we get $1127$, but $1127$ is prime If we add 2048, we get $1335$, but $1335$ is prime If we add 4096, we get $1544$, but $1544$ is prime If we add 8192, we get $1763$, but $1763$ is prime If we add 16384, we get $2401$, but $2401$ is prime If we add 32768, we get $2777$, but $2777$ is prime Therefore, the smallest integer $N$ is $\boxed{2777}$. The answer is: 2777

3 Inaccurate extraction of question information:
 Question: John drives for 3 hours at a speed of 60 mph and then turns around because he realizes he forgot something very important at home. He tries to get home in 4 hours but spends the first 2 hours in standstill traffic. He spends the next half-hour driving at a speed of 30mph, before being able to drive the remaining time of the 4 hours going at 80 mph. How far is he from home at the end of those 4 hours? He was driving at 60mph for 3 hours so he traveled 60*3=<<60*3=180>>180 miles. He was driving for 4 hours and spent 2 hours standing still so he drove 4-2=<<4-2=2>>2 hours. He drove 2 hours at 30mph so he traveled 2*30=<<2*30=60>>60 miles. <- should be half hour He drove 2 hours at 80mph so he traveled 2*80=<<2*80=160>>160 miles. <- should be 1.5 hour He drove 180 miles at 60mph, 60 miles at 30mph and 160 miles at 80mph for a total of 180+60+160=<<180+60+160=300>>300 miles. <- probably because 180-60-160 does not make sense so the model change it to a sum The answer is: 300

4 Wrong application of math knowledge: it's not a geometric sequence: Question: An airplane climbs 100 feet during the first second after takeoff. In each succeeding second it climbs 100 feet more than it climbed during the previous second. How many seconds does it take for the plane to reach an altitude of 12,000 feet above its takeoff height? The number of feet the plane is from the ground is the target of a geometric sequence with first term 0 and common ratio 2 We are asked to find the number of terms of this sequence which have a sum of 12000 The first term is 0, and the common ratio tells us that the second term is 2 more than the first, the third is 2 more than the second, and so on This means that the product of all the terms is $2^0\cdot{0}\cdot{2}\cdot{4}\cdot{8}\cdot{16}\cdot{32}\cdot{64}\cdot{128}\cdot{256}\cdot{512}\cdot{1024}\cdot{2048}\cdot{4096}\cdot{8192}\cdot{16384}\cdot{32768}\cdot{65536}\cdot{131072}\cdot{262144}\cdot{524288}\cdot{1048576}$, and the sum of all these terms is $262144 + 262145 + \cdots + 1048576 = 2\cdot{2^0} + 2\cdot{2^1} + \cdots + 2\cdot{2^{63}} = 2\left(1 + 2 + \cdots + 2^{63}\right)$ $2^{63}$ is equal to $2^{64} - 1$, so we add $1$ to the right hand side to get $2(1 + 2 + \cdots + 2^{64} - 1) = 2\left(2^{64} - 1\right) = 2^{65} - 1$ Finally, we subtract 1 from the right hand side to get $2^{65} - 1 - 1 = 2^{65} - 2^{65} = 1$, so it takes $\boxed{65}$ seconds for the plane to reach an altitude of 12,000 feet. The answer is: 65

5 Wrong logic/unreasonable step: does not make sense to use 6!-4!: Question: In how many ways can we seat 6 people at a round table if John and Sam insist on sitting next to each other? (Two seatings are

equivalent if one is a rotation of the other.) The number of all
seating arrangements is $6!$ The number of seating arrangements in
which John and Sam are next to each other is $4!$ Therefore, there
are $6!-4!=\boxed{75}$ different ways to seat the group. The answer
is: 75 Question: If $2ab = 12$, evaluate $8a^2b^2$. We have $2ab = 12
= 2^3 \cdot b^2$, so $8a^2b^2 = 2^3 \cdot (8a^2)b^2 = 2^3 \cdot 2b^2
= \boxed{8b^2}$. The answer is: $8b^2$

Format the output as follows: ####Type: [only the index of the error type
1-5] ####Explanation: [short and precise explanation].

Here is an example to analyze:

####Question: ...

####Generated Answer: ...

####GT reasoning: ...

Your response:

