# OpenReview forum: "Guiding Language Models Reasoning with Planning Tokens"
_ICLR.cc/2024/Conference — Submitted to ICLR 2024_

### Official Review · Reviewer_UavC · 2023-10-29

**Soundness:** 3 good
**Presentation:** 2 fair
**Contribution:** 2 fair
**Rating:** 5
**Confidence:** 4

**Summary:**

This paper uses planning tokens to guide large language models for math reasoning problems.  The planning tokens are discrete latent variables, which are append to each reasoning step generated by chain-of-thoughts reasonings.  The authors explore both K-means and Soft-VAE based clustering methods to generate the planning tokens for each reasoning steps.  Empirical experiments on three math reasoning datasets with three language models show the proposed method is effective.

**Strengths:**

1. The proposed method is attractive and novel due to the combination with discrete latent variable models with large language models.
2. The empirical improvements of LLaMa2 (13B) achieved by this method seem to be significant.

**Weaknesses:**

The main method is similar to prefix tuning.  However, there lacks of a strong baselines compared to prefix tuning.  For example, when the cluster number is equal to 5,  simply appending 5 trainable tokens in the front of each question should be a simple prefix-tuning baseline.  What are the performances?

2. All the experiments are based on human annotated reasoning steps on math problems.  It is unknown that how well the proposed method can generalize for automatically produced reasoning steps for general reasoning problems.  Since the title is not specific to math problems,  in order to achieve the wide-range claim in the title, it is necessary to investigate more general reasoning problems.

3.  When doing cluster for each reasoning step, the contextual information of each step seems not explored.  In section 2.3.2, to obtain the representation vector, each reasoning step is separately encoded.  The authors should consider a contextual version, which use the prefix reasoning steps and the question context to generate the representation vectors of the current step.
4. Soft-VAE is not novel.  Why not simply using VQ-VAE?
5. Although in the introduction, one important motivation is to address the issues of the reasoning steps progressively drifting away from the correct reasoning flow,  there are on such designs or reasoning in later sections of why the planned token helps on this issue.  In my view,  a hierarchical generation might be better correlated with this motivation.  First, given the question context.   Second, we tune the language models to generate the planning tokens.   Third, and then based on the question context and the planning tokens, for each reasoning step, we first copy a specific planning token generated from  the second step, and then output the reasoning process.  But in this method, the second step is missing, which I therefore hypothesize that there lacks a correspondance between the proposed method and the motivation.
6. The analysis of the planning tokens should be enriched. Currently, the analysis is quite weak and not enough.

**Questions:**

1. Do you do the clustering based on the training set or just do the clustering using the test set?
2. In Section 3.1, why do you subsample 1K test samples from GSM8K and MATH?  Why not using all the test sets?
3. The optimal setting of the number of cluster can be varied across datasets. It is better to also perform ablation studies on the other dataset.

Minors:
In Table 2, the dataset (GSM8K) and the model (LLaMA2 7B) should be explicitly described.

---

> ### Author Response · Authors · 2023-11-15
>
> Thank you for your detailed review. Below is our response to the weakness section:
>
> 1. **Compare to prefix tuning**: Please refer to the general response. We have tried the prefix tuning baseline in the early stage of the project; they perform significantly worse than our current baselines: N/A (full fine-tuning/LORA), and General (adding the same prefix tokens in front of each step). We will include these results in our main table.
> 2. **Title**: While our proposed method has great potential on applying to various kinds of reasoning, our empirical results are currently focused on the math word problem datasets. We will restrict our title to math reasoning by changing it to: Guiding Language Models Math Reasoning with Planning Tokens
> 3. **Contextual information of each step**: We actually have already taken the contextual information into consideration, by using the contextualized embedding of each reasoning step produced by the base LLM. More specifically, we concatenate the question and reasoning steps, and encode the whole sequence with the base LLM. We then average the contextual embeddings of all tokens in a reasoning step. This makes the embedding of each reasoning step conditioned on the previous ones. We will clarify this important detail in the paper.
> 4. **VQ-VAE v.s. SQ-VAE**: We actually tried VQ-VAE first, and found it is very unstable to train on the step embeddings, so we switched to a soft version of it. We agree that this soft version of VQ-VAE is not a significantly novel contribution. Instead, this is an implementation choice for our proposed method.
> 5. **Hierarchical generation**: Thanks for the suggestion. We tried to adopt a sparse attention scheme in the earlier stage of the project to enforce a stronger dependency constraint between the planning tokens and the reasoning step that follows, similarly to what you are suggesting. More specifically, we modified the attention mask to make the generated step only depend on the corresponding planning tokens and the planning tokens to only depend on the previous planning tokens and the question, however the performance dropped quite dramatically, showing that the flexibility induced by full attention is important for final performance. Also, note that hierarchical generation usually comes at the cost of more difficult parallelization, given that two LLMs (planner and generator) might need to be kept in memory. All in all, these are interesting avenues to be explored, but the simplicity and effectiveness of our approach remain still attractive in our opinion.
> 6. **Analysis of planning tokens**: We kindly ask the reviewer to provide some more details on which kind of analysis is missing. In section 3.3, we first provide an analysis of the error rate versus the reasoning length. Then we provide a detailed error taxonomy. Finally, we provide a probing-based analysis of the inference planning token types on page 8, titled “Distinguishability of the Induced Clusterings”.
>
> Below is our response to the question section:
>
> 1. **Clustering**: We only do clustering on the training set. The testing set is not used in our learning process.
> 2. **Testing set subsampling**: Because at testing time, the model will need to generate a long sequence of text as the solution, which makes testing slow. Considering our time and computing constraints, we decided to restrict the size of our testing sets to speed up our experiments. Thank you for pointing this out. We will clarify this experimental choice in the paper.
> 3. **Ablation on all datasets**: We would love to provide them on all datasets, but we weren’t able to do so because of our time and compute constraints, since these experiments are expensive to run.

---

> > ### Author Response · Authors · 2023-11-22
> >
> > Dear reviewer, it's the last day of the discussion period. We are wondering if our rebuttal has addressed your concerns and if you would like to reconsider your score. Thank you.

---

> > > ### Comment · Reviewer_UavC · 2023-11-23
> > >
> > > I have read all the rebuttal and the new version.
> > >
> > > 1. Why the tunable parameters of Prefix and SQ-VAE are different?  If the number of the planned tokens are the same, the tunable parameters should be the same.
> > >
> > > 2. For the analysis,  I did not feel the current analysis establish a good connection to why the induced token help on reasoning.  On one side, for different contexts,  if the induced tokens are the same,  do they belong to the similar problems?  On the other side,   if the thoughts belongs to the same pattern for two different problems,  are they intuitive enough for us to interpret we can solve similar problems using the same strategy.  At least some pattern analysis instead of single induced cluster should be reported.
> > >
> > > Currently,  I will keep my original score but if the prefix tuning setting is no problem,  I will improve it to 6.  Thanks.

---

> > > > ### Author Response · Authors · 2023-11-23
> > > >
> > > > We are happy that our response clarified some of your concerns and thanks for considering raising the score. We want to further clarify a few things here:
> > > > 1. For the new **Prompt** and **Prefix** baselines, we stick to the original implementation of prompt tuning and prefix tuning. We increased the number of Prefix and Prompt embeddings to match the parameters of our SQ-VAE approach (which tuned both planning token embeddings and LORA adapters) but this did not improve performance (10.2% for prompt tuning on GSM8K with Llama 7B). A more expressive baseline that can be viewed as an advanced version of **Prompt** tuning is our **General** baseline, which also trains both general token embeddings and LORA adapters, but still underperforms our SQ-VAE approach, albeit using the same # of parameters.
> > > > 2. We agree with the reviewer that there can be better ways to interpret the learned planning token types. We did not find a satisfying interpretation of the clustering and thus we resorted to a probing-based analysis. We have thought about querying a pre-trained LLM, such as  GPT4, with the centroid tokens; however, the whole dataset seems to be too large to be fed into GPT4. We agree the pattern analysis proposed by the reviewer is interesting, thanks for suggesting it. We will try to add an analysis and investigate whether similar planning token patterns could cluster similar problems together.

---

> ### Comment · Reviewer_UavC · 2023-11-23
>
> Did you try prefix tuning with LoRa? In this setting, for prefix tuning the tunable tokens are randomly initialized and do not represent an explicit cluster.  For SQ-VAE, the planned tokens are induced by your method. This is the only difference. LoRA is kept the same.
>  In this way, the number of tunable parameters should be the same.

---

### Official Review · Reviewer_9J4w · 2023-10-29

**Soundness:** 4 excellent
**Presentation:** 4 excellent
**Contribution:** 4 excellent
**Rating:** 8
**Confidence:** 3

**Summary:**

To tackle the issue of lack of consistency among reasoning steps in solving math word problems, this paper introduces planning tokens to help with ‘global’ reasoning. Clustering  (Soft Q-VAE) is used to learn the planning tokens along with existing reasoning datasets and parameter efficient fine-tuning. Contributions include the different clustering methods and a detailed error analysis.

**Strengths:**

- Originality: Most existing work on improving reasoning focuses on better prompting techniques which makes the use of planning tokens more original.
- Quality: The method development and experimental results are thorough.
- Clarity: The paper was easy to understand.
- Significance: Similar to originality, the planing tokens can be widely incorporated into many other ideas.

**Weaknesses:**

- The training process relies heavily on the annotated reasons which limit the flexibility of the model.
- The performance on the MATH dataset is notably lower than the others.

**Questions:**

- Do the mistakes from the MATH dataset fall under the same error taxonomy (from section 3.3) as the GSM8K and Aqua datasets?

---

> ### Author Response · Authors · 2023-11-15
>
> Thank you for your positive review!
>
> The MATH dataset is significantly more difficult compared to the other two datasets. It remains challenging for very strong LLMs, such as GPT-4.
>
> * Data source: The MATH dataset comes from the AMC math competitions. In contrast, GSM8K comes from middle school and high school math exams; AQUA comes from multiple choice GRE math tests.
> * Answer format: The answers of MATH data are free-form text, e.g., Latex expressions such as \frac{1-\sqrt{2}}{2}, which makes it hard for the model to get every bit of the answer correctly. In contrast, the answers of GSM8K are single int numbers; the answers of AQUA are single characters, indicating one of the choices.
>
> When we summarize the error taxonomy, we take all three datasets into consideration. However, most of the mistakes in the MATH dataset fall in the wrong logic category, as the problems are not that obvious to approach, and the model seems to be confused about what the next step should be. Thanks for pointing this out. We will include an error taxonomy of MATH in the paper.

---

### Official Review · Reviewer_C3Cd · 2023-10-31

**Soundness:** 1 poor
**Presentation:** 2 fair
**Contribution:** 1 poor
**Rating:** 1
**Confidence:** 4

**Summary:**

The paper proposes to augment "planning tokens" in reasoning text in fine-tuning LLM and update the embeddings of "planning tokens" together with other params.

Experiments show that this trick has positive effects.

**Strengths:**

The paper tries multiple ways to infer latent planning token types.

The experiment results are solid.

**Weaknesses:**

The significance of this paper is low because tuning augmented tokens is not a new thing in NLP and the empirical findings in this paper are not interesting conditioned on previous work. I personally learned nothing from the paper.

The paper cites no previous work that learns augmented tokens.

Related references are:

Li and Liang, 2021 Prefix-Tuning
Qin and Eisner, 2021 Learning how to ask

**Questions:**

NA

---

> ### Author Response · Authors · 2023-11-15
>
> We thank the reviewer for pointing out the related works that we missed; we will adjust the text accordingly.
>
> We emphasize that we are not claiming that tuning augmented tokens is a contribution of this work. Many existing works on prompt/prefix tuning [1-3] served as inspiration for our method. The main contribution of our work lies in a particular application of soft tokens for significantly improving the reasoning ability of modern LLMs.
>
> Many differences distinguish our work from [2] and [3]:
>
> * **Latent token type**: All previous approaches assume the ground-truth information about the token type to be given both for training and, crucially, testing. For example, [3] assumes knowledge of the relation type. In our work, we explore a more general setting, in which the token type is a latent variable and we train the reasoning model to directly predict the planning token at inference time. Therefore, the model needs to infer the planning token type from the previous reasoning steps. In one of our baseline settings, we use `Arithmetic` planning tokens, which is a particular case of token type inference informed by prior knowledge: the arithmetic operation can be understood as a relation type that links inputs and reasoning steps. Empirical results (Table 1) show that the planning tokens inferred by K-Means and SQ-VAE always provide better performance than the Arithmetic planning tokens, which indicate the importance of a learning-based token type inference algorithm.
> * **Fixed token location**: Our planning tokens do not have a fixed location in the input prompt as in [1-3]. Instead, they will be freely generated by the model during the inference time, instead of being fed as input prompts. We only provide those inferred planning tokens in the training data, and they are added at multiple places of the sequence, instead of only in the front or in the middle.
> * **Integrability**: Our method is supposed to be integrated into other training/fine-tuning paradigms (e.g. full fine-tuning or LORA) to improve LLM’s math reasoning capacity, instead of using on its own.
> * **Harder task**: In this work, we test the proposed methods on a set of math reasoning tasks, which are arguably more difficult than the relation completion tasks used in [3]. Our method is designed to specifically enhance the reasoning abilities of modern decoder-only LLMs. This is in contrast to [3], which is built for relation completion tasks in the context of masked LMs. As shown in the general response, simple prompt/prefix tuning systems underperform in our setting.
>
> To address the reviewer's statement on novelty, we reiterate our contributions which distinguish us from the related work raised in the review:
>
> * As shown in our general response, traditional prompt/prefix tuning techniques do not work well on hard math reasoning tasks, our method redesigned the old paradigm and turned it into a strong auxiliary technique to improve the math reasoning capabilities of other fine-tuning paradigms.
> * Planning tokens improve the intra-step consistency of the generated solution, and increase the model’s reasoning capacity by specializing it to different reasoning types.
> * Our method is able to infer the type of latent planning tokens instead of using fixed, predefined types.
>
> [1] Lester, Brian, Rami Al-Rfou, and Noah Constant. "The Power of Scale for Parameter-Efficient Prompt Tuning." Proceedings of the 2021 Conference on Empirical Methods in Natural Language Processing. 2021.
>
> [2] Li, Xiang Lisa, and Percy Liang. "Prefix-Tuning: Optimizing Continuous Prompts for Generation." Proceedings of the 59th Annual Meeting of the Association for Computational Linguistics and the 11th International Joint Conference on Natural Language Processing (Volume 1: Long Papers). 2021.
>
> [3] Qin, Guanghui, and Jason Eisner. "Learning How to Ask: Querying LMs with Mixtures of Soft Prompts." Proceedings of the 2021 Conference of the North American Chapter of the Association for Computational Linguistics: Human Language Technologies (NAACL-HLT). 2021.

---

> > ### Author Response · Authors · 2023-11-22
> >
> > Dear reviewer, it's the last day of the discussion period. We are wondering if our rebuttal has addressed your concerns and if you would like to reconsider your score. Thank you.

---

> > > ### Comment · Reviewer_C3Cd · 2023-11-22
> > >
> > > I appreciate your clarification and new results. But I am not convinced of the contribution of this work and let me try to explain.
> > >
> > > > The main contribution of our work lies in a particular application of soft tokens for significantly improving the reasoning ability of modern LLMs.
> > >
> > > First, application of soft tokens to a specific reasoning task is not a technical innovation by itself, unless well-suited architectural or methodological changes have been made for this application.
> > >
> > > Second, repitching this paper needs significant structural revisions of the paper, in parallel to the local clarifying edits that you have already done (thank you for doing it btw).
> > >
> > > > ground-truth information about the token type to be given both for training and, crucially, testing.
> > >
> > > I doubt this.
> > >
> > > The idea of using soft tokens is to tune input embeddings of some new token types, which may or may not be initialized with existing tokens. In [3], some tokens are initialized in sophisticated ways, but it doesn't mean that their token types are restricted, does it?
> > >
> > > > they will be freely generated by the model during the inference time, instead of being fed as input prompts. We only provide those inferred planning tokens in the training data, and they are added at multiple places of the sequence, instead of only in the front or in the middle.
> > >
> > > I agree that this is a new technique, for which I would like to consider raising my score.
> > >
> > > But I am afraid this technique is still in a broken / limited form.
> > >
> > > Correct me if I am wrong: do you learn to infer the position of latent tokens during training? If not, then inference will mostly mimic what the LM has seen during training, right? Then what drives the generation of latent tokens is still where you choose to place them in training data, right?
> > >
> > > In addition, as I understand of [1]-[3], "prefix-tuning" is just a name and soft tokens are also added somewhere else but not only at the beginning of the prompts.
> > >
> > > > improve LLM’s math reasoning capacity,
> > >
> > > Why math specifically but not other kinds of reasoning? Why not language modeling in general? The argument for math seems arbitrary to me.

---

> > > > ### Author Response · Authors · 2023-11-23
> > > >
> > > > We are happy that our response clarified some of your concerns and thanks for considering raising the score. We want to further clarify a few things here:
> > > >
> > > > 1. **Novelty/contribution**: our proposed method is inspired by the previous soft-token tuning methods, but is significantly different from them. As we showed in the main table, traditional prompt/prefix tuning does not work on math reasoning tasks.
> > > >
> > > >     In our opinion, our method comprises significant architectural and methodological changes w.r.t. soft prompts and prefix-tuning.
> > > >     We would like to reiterate those:
> > > >     * We propose to make the LM generate the soft-prompts (i.e. the planning tokens) such that they can guide the model at the beginning of each chain-of-thought (CoT) step. We kindly refer the reviewer to section 2.1 for a Bayesian interpretation of this framework. In short, the planning token can be viewed as a discrete proxy of the underlying latent variable governing the generation of the whole sequence, thus providing the model a better control.
> > > >     * The planning tokens can be inferred from the training data in an unsupervised manner, instead of being assigned based on the task that they are intended to perform.
> > > >     * Planning tokens can be easily integrated into other (parameter-efficient) fine-tuning techniques.
> > > >
> > > > 2. **Ground-truth information of token types**: We believe there are some misunderstandings here, we do not mean the initialization of the token embeddings. In [1-3], the set of tokens to be used is tied to the specific task or relation to be tackled by the model. In contrast, in our system, we do not assume any prior knowledge of the ground-truth reasoning types when assigning planning tokens to different reasoning steps. In fact, in one of our baseline settings, we tried to use arithmetic operations as such ground truth, but as reported in Table 1, our unsupervised clustering-based planning token inference methods work significantly better. This is one of our main findings.
> > > >
> > > > 3. **Inference of latent token position during training**: In this work, we assume that planning tokens occur before each reasoning step in the training data, therefore we do not infer the “best” position of the latent tokens during training. Relying on a heuristic for the position of the latent tokens is a simplifying assumption that can be relaxed in future work. In our opinion, this doesn’t make our method “broken”, given that it performs well empirically. Note that a similar heuristic (Arithmetic) for inferring the token type does *not* work well and strengthen the novelty of our approach. To see what inspired us to put planning tokens at the beginning of each step, please see the beginning of section 2.1. Even if we rely on a position heuristic, during testing, our model is able to decide where to generate the planning tokens by itself. We are not aware of any prior work that generates such tokens freely by an LM. We cannot guarantee that the model will generate the planning tokens at the beginning of each step at testing time, but the model is able to learn from the training data that the planning tokens are supposed to be placed in between each step. The fact that the LM can do that is not known/shown by [1-3] or other existing soft-prompt tuning works, we believe this is also an important feature that distinguishes our work from prior works.
> > > >
> > > > 4. **Beyond math reasoning**: the reviewer is correct, it is exciting that this framework can be applied to any text generation task. Our hunch is that we expect it to especially improve for long-form generation. We think this is a very exciting direction for future work. For the purpose of this paper, we wanted to analyze this for math reasoning because 1) they require multi-step reasoning; 2) there exists large-scale human-annotated reasoning chain data in natural language; and 3) to be comparable with recent works on LLM reasoning.

---

### Official Review · Reviewer_1L1z · 2023-10-31

**Soundness:** 2 fair
**Presentation:** 4 excellent
**Contribution:** 3 good
**Rating:** 6
**Confidence:** 3

**Summary:**

This paper studies the way of using planning tokens to improve LLMs on reasoning consistency. Experiment results show that the proposed method outperforms finetuning LLMs directly. Analysis results provide intuitions to show how the proposed method works.

**Strengths:**

Generally, this is a good paper. The paper is well-written, and the idea is quite interesting. The proposed method is novel and comprehensive experiments are done to prove that the method works.

**Weaknesses:**

The proposed method is similar to prefix-tuning. This could be added as a baseline for comparison. Besides, there are some prompt-based methods targeted for reasoning consistency, which can also be added in experiments. Only comparing with full finetuning may not be enough.

**Questions:**

See weakness

---

> ### Author Response · Authors · 2023-11-15
>
> Thank you for your positive review. We have tried prefix-tuning baselines in the early stage of the project; they were significantly worse than our current baselines: N/A (full fine-tuning/LORA), and General (adding the same prefix tokens in front of each step). Please refer also to the general response. Thanks for pointing this out. We will update these results in our result section and add discussion.
>
> We kindly ask the reviewer to further clarify which prompt-based methods they referred to as our potential baselines. We are happy to add them!
>
> [1] Lester, Brian, Rami Al-Rfou, and Noah Constant. "The Power of Scale for Parameter-Efficient Prompt Tuning." Proceedings of the 2021 Conference on Empirical Methods in Natural Language Processing. 2021.
>
> [2] Li, Xiang Lisa, and Percy Liang. "Prefix-Tuning: Optimizing Continuous Prompts for Generation." Proceedings of the 59th Annual Meeting of the Association for Computational Linguistics and the 11th International Joint Conference on Natural Language Processing (Volume 1: Long Papers). 2021.

---

### Author Response · Authors · 2023-11-15
**General response**

We want to thank all reviewers for their insightful comments. We want to first clarify a few points raised by the reviewers and initiate the discussion.

One of the common concerns is about the prefix-tuning [2] / prompt-tuning [1] baselines: we would like to point out that our General baseline can be seen as a more powerful version of prompt-tuning [1] that adds learnable tokens between each reasoning step instead of just before the input. With the same number of tokens (6) used in the main table (Table 1) on GSM8K with Llama 2 (7B),

* the result of only tuning the embeddings of the prefix tokens inserted at the very beginning (i.e. standard prompt-tuning [1]) was 15.2%.
* The result of tuning the embedding of the prefix tokens at all layers (i.e. prefix-tuning [2]) was 8.9%.
* The result of tuning the embeddings of the same prefix tokens inserted at the beginning of each reasoning step (i.e. our General baseline without LORA) was 16.0%.
* They are significantly worse than our current baselines: N/A (i.e. LORA, 38.2%), and General (i.e. inserting the same prefix tokens in front of each step along with LORA, 38.5%).

We will include the complete results on other datasets in our main table and discuss the connection between this line of work and our work.

We will upload a new version of our paper as soon as we finish all the modifications listed below:

1. Change the title to: Guiding Language Models Math Reasoning with Planning Tokens
2. Add prompt-tuning and prefix-tuning baselines to the main table.
3. Discuss the line of work on soft-prompt tuning in the related work section.
4. Add error taxonomy on MATH.
5. Clarification of implementation and experimental choices: contextualized embedding, SQ-VAE v.s. VQ-VAE, testing set subsampling.

[1] Lester, Brian, Rami Al-Rfou, and Noah Constant. "The Power of Scale for Parameter-Efficient Prompt Tuning." Proceedings of the 2021 Conference on Empirical Methods in Natural Language Processing. 2021.

[2] Li, Xiang Lisa, and Percy Liang. "Prefix-Tuning: Optimizing Continuous Prompts for Generation." Proceedings of the 59th Annual Meeting of the Association for Computational Linguistics and the 11th International Joint Conference on Natural Language Processing (Volume 1: Long Papers). 2021.

---

> ### Author Response · Authors · 2023-11-21
> **Updates**
>
> Dear reviewers, we have uploaded the revised version of our paper. We apologize for the wait as the Llama 13B model baselines are very slow to run. Below is the list of modifications (marked in blue in the paper draft):
>
> 1. Our title is changed to Guiding Language Models Math Reasoning with Planning Tokens.
> 2. We added a more in-depth discussion on the Bayesian motivation behind our method, which we hope will make our proposed planning token method better motivated.
> 3. We clarified that we use contextualized embedding of a step that encodes the information from the question and all previous steps.
> 4. We clarified the mathematical guarantees and motivations of using a soft-quantized VAE to infer the planning token types. We also clarified that we didn’t use VQ-VAE because it’s hard to train on our data.
> 5. We have added prompt-tuning and prefix-tuning baselines to the main table, which are 10+% lower than our original baselines.
> 6. We revised the error taxonomy section by manually verifying the GPT4-produced evaluations and added error taxonomy on both MATH and GSM8K.
> 7. We clarified that we subsample the testing set for time efficiency.
> 8. We discussed the soft-prompt tuning works in related work.
>
> We want to thank you for taking the time to review our paper. We know it’s a busy moment, but we want to remind the reviewers that the discussion period is ending soon. We are happy to address any remaining concerns before the discussion period ends.

---

> > ### Author Response · Authors · 2023-11-21
> >
> > Dear Reviewers,
> >
> > Thank you for your time. We know it’s a busy moment, but we would like to know if there’s anything else we can do to clarify your concerns about our paper.
> >
> > Please see the summary of the paper revision above.
> >
> > ***Reviewers 1L1z, 9J4w:***
> >
> > Thank you for your positive feedback, we hope the rebuttal helped in answering your questions. Please let us know if you have other comments and concerns.
> >
> > ***Reviewer C3Cd:***
> >
> > Your main concern was our missing reference and the lack of comparison with a few recent works, which together make our work less novel.
> >
> > We thank you for pointing out the missing reference. We have discussed the connection and difference between our work and prior works both in the response and the revision of our submission. Please find in the revision, we report new experimental results directly comparing our system with prior works such as Prefix-Tuning as you suggested. In the response, we have also emphasized our novelty, we hope this helps clarify your concerns.
> >
> > Is there anything else that we can do to address your concerns?
> >
> > ***Reviewer UavC:***
> >
> > Your main concern was about comparison with some recent works such as Prefix-Tuning. You have also asked some clarifying questions regarding the motivation and analysis sections.
> >
> > In the response and in the revision, we have discussed the connection and difference between our work and prior works. Specifically, we have included baselines such as Prefix-Tuning in the main table. Once again, in both the response and the revision, we have addressed your concern about motivation. To make our contribution clearer, we have also modified the title.
> >
> > With respect to additional analysis you asked in the review, we kindly ask you to clarify on what specific analysis do you expect that can make our work stronger. We are happy to include them in the next version.
> >
> > Is there anything else that we can do to address your concerns?
> >
> > Thank you.

---

### Meta-Review · Area_Chair_GEAW · 2023-12-07

**Metareview:**

This paper uses planning tokens as a guiding mechanism for large language models in the context of math reasoning problems. These planning tokens are discrete latent variables that are appended to each reasoning step generated through chain-of-thought reasoning. The authors explore two approaches, the K-means and Soft-VAE-based clustering methods, to generate these planning tokens for each reasoning step. Empirical experiments conducted on three different math reasoning datasets employing three distinct language models indicate the effectiveness of the proposed approach.

The author rebuttal has addressed some of the raised concerns. However, there remain some significant weaknesses as pointed out by the reviewers:

- Lack of Novelty: The paper's utilization of soft tokens and the idea of using multiple soft tokens are not novel concepts. As a result, the primary novelty lies in the VAE-based approach to learn planning tokens from data. Unfortunately, the paper does not convincingly demonstrate a clear performance advantage of this method over the K-Means approach.
- Insufficient Baseline Comparison: The experiments conducted in the paper do not adequately demonstrate the superiority of the learned soft tokens. Additionally, as highlighted by reviewer 9J4w, the method for learning tokens relies heavily on the specific characteristics of the dataset annotations. Moreover, the newly added prefix and prompt baselines are not trained concurrently with LoRA, creating an unfair comparison.

Furthermore, we had a discussion with the reviewer who gave it a score of 8. The reviewer shares concerns regarding the experimental comparison and opted not to advocate for the paper in its present state.

**Justification For Why Not Higher Score:**

Considering the lack of novelty and the insufficient baseline comparison raised in reviews and the meta-review.

**Justification For Why Not Lower Score:**

NA

---

### Decision · Program_Chairs · 2024-01-16

Reject